# First Case Report of Choledochoenterostomy in a Cat with Biliary Obstruction Due to Cholangiohepatitis and Papillary Stenosis

**DOI:** 10.3390/ani15172634

**Published:** 2025-09-08

**Authors:** Nicole Diana Wolf, Juliette Bénédicte Burg-Personnaz, Jennifer Stéphanie Eiermann, Simona Vincenti

**Affiliations:** 1Division of Small Animal Surgery, Department of Clinical Veterinary Medicine, Vetsuisse-Faculty, University of Bern, 3012 Bern, Switzerland; 2Division of Clinical Radiology, Department of Clinical Veterinary Medicine, Vetsuisse-Faculty, University of Bern, 3012 Bern, Switzerland; juliette.burg@unibe.ch; 3Division of Small Animal Internal Medicine, Department of Clinical Veterinary Medicine, Vetsuisse-Faculty, University of Bern, 3012 Bern, Switzerland; jennifer.eiermann@unibe.ch

**Keywords:** choledochoenterostomy, extrahepatic biliary tract surgery in felines, cat, cholangiohepatitis, dilation of the common bile duct, case report

## Abstract

A 2-year-old neutered male cat was brought in with a year-long history of poor appetite, weight loss, vomiting, and increased drinking and urination. On exam, the cat was very thin and jaundiced. Lab tests showed abnormalities in liver values, anemia, and low vitamin B12. Imaging revealed a marked dilation of the common bile duct (CBD) and gallbladder, along with signs of pancreatic involvement. Tests confirmed a bacterial liver infection caused by *Escherichia coli* and *Peptostreptococcus canis*. A cholecystectomy, choledochoenterostomy, and placement of a cholecystostomy tube as well as an abdominal drain and jejunal feeding tube were performed. Although the cat initially improved, it later developed sepsis and died suddenly 10 days after surgery due to unknown causes. This case demonstrates the complexity of diagnosing and treating severe liver and bile duct disease in cats and reports the first-known use of a specific bile-duct bypass surgery (choledochoenterostomy) in this species.

## 1. Introduction

Surgery on the extrahepatic biliary system in cats is indicated due to extrahepatic biliary obstruction, which may occur extraluminally due to inflammation like pancreatitis or neoplasia of nearby organs or intraluminally because of cholelithiasis, biliary mucocele, biliary neoplasia, infection, or trauma [1]. In most cases, timing of surgery is controversial, and medical management is the first choice. Suggestions to convert to surgery are a worsening hyperbilirubinemia and progressive distension of the common bile duct over a 7- to 10-day period [2,3].

Common surgical procedures on the extrahepatic biliary system in cats are biliary stenting or placing of a cholecystostomy tube if a reversible disease process like pancreatitis is anticipated [1,4]. Cholecystotomy or choledochotomy for removal of choleliths is rarely indicated, but if the patency of the common bile duct (CBD) is present, cholecystectomy is preferred to additionally prevent recurrence in cases of cholelithiasis. Furthermore, cholecystectomy is performed in the event of biliary mucocele, trauma, or neoplasia. If the patency of the CBD cannot be demonstrated but the gallbladder is vital, a cholecystoenterostomy should be considered. Choledochoenterostomy is routinely performed in human medicine if the gallbladder is necrotizing, but to the authors’ knowledge, it has only anecdotally been described in cats due to the small size of the CBD (only 2 to 4 mm in diameter) [4].

The present case report describes the surgical procedure of a choledochoenterostomy in combination with cholecystectomy and choledochostomy tube placement after dilation of the CBD due to bacterial cholangiohepatitis and stenosis of the papilla major in a domestic shorthair cat.

## 2. Case Description

### 2.1. Signalment and Patient History

A 2-year-old male neutered domestic shorthair cat was presented to the Small Animal Clinic of Bern due to a 12-month history of anorexia, apathy, weight loss, vomiting, and polyuria/polydipsia (PU/PD).

The initial workup at the first-opinion practice included clinical examination in combination with blood work, which resulted in the suspicion of cholangiohepatitis and chronic pancreatitis. The tests for FeLV/FIV were negative. Treatment included Ursodeoxycholic acid (Ursochol, Zambon Ltd., Cadempino, Switzerland) 5 mg/kg per oral (PO) once daily (SID) and Amoxicillin-Clavulanic acid (Cylanic, Livisto, Barcelona, Spain) 12.5 mg/kg PO twice daily (BID). The antibiotic treatment was discontinued more than 6 months prior to administration to our institution. The cat was referred for further diagnostic workup and treatment to the Small Animal Clinic of Bern. The owners signed an informed consent for the entire procedure.

### 2.2. Physical Examination Workup on First Presentation

On admission to the emergency service, the cat was in lateral recumbency but responsive with a heart rate of 120/min, respiratory rate of 32/min, temperature of 37.7 °C, icteric mucous membranes, and a shaggy coat. The weight was 2.2 kg with a BCS of 2/9.

### 2.3. Laboratory Work

The hematology results showed a mild macrocytic normochromic non-regenerative anemia (26%) and a regenerative left shift (0.88 × 10^9^/L). The chemistry panel (Table 1) showed a mild hyperproteinemia (90.8 g/L) with hypoalbuminemia (25.4 g/L) and hyperglobulinemia (65.4 g/L), a mild hyperbilirubinemia (3.7 umol/L), elevated liver enzymes (ALT 300 U/L, AST 70 U/L, gGT 17 U/L, GLDH 44 U/L), and elevated lipase (60 U/L). The serum amyloid A was within normal limits. Serum cobalamin concentration was severely diminished with 122.10 pg/mL.

### 2.4. Diagnostic Imaging

Ultrasonographic evaluation revealed marked dilation of the gallbladder and CBD (Figure 1). Both contained a marked amount of gravity-dependent echoic material with hyperechoic foci and speckles, alongside non-gravity-dependent anechoic material. The walls of the gallbladder and CBD were moderately thickened and hyperechoic, and they exhibited mild irregularity.

The CBD demonstrated abrupt luminal tapering proximal to the major duodenal papilla (Figure 1B), while the papilla itself appeared within normal limits sonographically. Moderate dilation of the intrahepatic bile ducts was noted. The pancreatic duct was mildly dilated at its most distal aspect (Figure 1B). The stomach was moderately displaced toward the right cranial abdomen, consistent with mass effect from the adjacent biliary structures. A mild amount of anechoic peritoneal effusion was identified.

The findings were consistent with an extrahepatic biliary obstruction, most likely at or just proximal to the major duodenal papilla, secondary to a mucous plug, stricture, or other causes of obstruction. Reactive peritoneal effusion and mild pancreatic ductal dilation may reflect early secondary inflammatory or pressure-related changes.

Ultrasound-guided fine-needle aspirates of the liver, bile, and peritoneal effusion were performed. Cytology was performed on all samples, and a bacterial culture was performed on the liver and bile samples.

### 2.5. Cytology and Bacteriology

Cytologically, the samples showed neutrophilic inflammation with a mixed bacterial infection in the bile, while bacteriological swabs of the liver and bile were markedly positive for hemolytic *Escherichia coli* as well as an anaerobic mixed flora (Thiogluconate Enrichment broth, 5% sheep blood agar, and aerobe and anaerobe incubation at 37 °C). Furthermore, the liver swab was positive for *Peptostreptococcus canis*. The identified bacteria were sensitive to all common antibiotics (minimal inhibitory concentration (MIC) values ranging between 0.25 and 20; MIC value for Ampicillin at 2).

Cytology of the free abdominal fluid revealed a mild neutrophilic inflammation without any bacterial growth.

### 2.6. Treatment and Plan

The results of the anamnesis, clinical exam, and further workup were indicative of a bacterial cholangiohepatitis, potentially secondary to a stenosis of the CBD proximal to the papilla.

The cat was hospitalized and treated for two days with intravenous fluids (Lactated Ringers Solution), Ondansetron (Ondansetron Labatec, Labatec Pharma SA, Meyrin, Switzerland) 0.3 mg/kg IV three times daily (TID), Buprenorphin (Bupaq P, Streuli Pharma, Uznach, Switzerland) 0.01 mg/kg IV TID, Ampicillin Sulbactam (Ampicillin-Sulbactam Kabi, Fresenius GmbH, Bad Homburg, Germany) 30 mg/kg IV TID, S-Adenosylmethionin-Silybin (Denamarin S, Nutramax Laboratories Inc., Lancaster, SC, USA) 1 pill PO once daily (SID), and Ursodeoxycholic acid (Ursochol, Zambon Ltd., Cadempino, Switzerland) 10 mg/kg PO SID. The clinical state improved, and the cat was discharged after two days with Amoxicillin-Clavulanic acid (Cylanic, Livisto, Barcelona, Spain) 12.5 mg/kg BID, Ursodeoxycholic acid (Ursochol, Zambon Ltd., Cadempino, Switzerland) 10 mg/kg po SID, S-Adenosylmethionin-Silybin (Denamarin S, Nutramax Laboratories Inc., Lancaster, SC, USA) 1 pill SID, and the recommendation to supplement Cyanocobalamin (Vitamin B12 Vitarubin, Streuli Pharma, Uznach, Switzerland) 250 µg once a week. The owners were advised to have liver values rechecked by their primary veterinarian the following week and to schedule a follow-up appointment at our institution in three weeks for full blood work and repeat abdominal ultrasound, provided the cat remained clinically stable.

### 2.7. Physical Examination and Work Up on Second Presentation (21 Days After First Presentation)

The cat’s general condition had slightly deteriorated with increased frequency of vomiting and reduced appetite. At presentation, the cat appeared moderately debilitated but responsive. Aside from persistent icteric mucous membranes, all vital parameters were within normal limits. Abdominal palpation revealed a marked non-painful enlargement in the cranial abdomen.

### 2.8. Laboratory Work

Hematology revealed a mild worsening of the macrocytic normochromic non-regenerative anemia (23%) but a resolution of the regenerative left shift.

The chemistry panel (Table 2) showed an unchanged hypoalbuminemia (23.7 g/L) and hyperglobulinemia (58.7 g/L). Furthermore, there was a markedly progressive hyperbilirubinemia (29.2 umol/L), and the liver enzymes (ALT 1140 U/L, AP 158 U/L, AST 7276 U/L, gGT 22 U/L, and GLDH 245 U/L) and lipase (111 U/L) also increased. The serum amyloid A rose from normal to 30.4 mg/L.

The full coagulation profile (Table 3) showed an increase of PTT to four times its reference (66.7 s (reference 11.9–16.2 s)) and an elevation of the D-Dimers.

### 2.9. Diagnostic Imaging

Findings were overall unchanged to mildly progressive, with the additional observation of a thickened major duodenal papilla (Figure 2).

### 2.10. Treatment and Plan

Due to the worsening of the general condition, deteriorating blood values, and persistently abnormal ultrasonographic findings, the decision was made to proceed with surgery the following day. To stabilize the cardiovascular system and address suspected coagulopathy, a fresh frozen plasma transfusion was administered pre- and intraoperatively with 10 mL/kg. Ampicillin-Sulbactam (Ampicillin-Sulbactam Kabi, Fresensius GmbH, Bad Homburg, Germany) was administered at 30 mg/kg IV preoperatively and continued postoperatively TID. Marbofloxacin (Marbocyl, Vetoquinol SA, Lure Cedex, France) was initiated intraoperatively at 4 mg/kg and continued SID postoperatively.

### 2.11. Explorative Laparotomy and Surgical Procedure

The cat was placed in dorsal recumbency, clipped, and aseptically prepared for surgery. Following clipping, the extent of the cranial abdominal enlargement was visible (Figure 3). Abdominal exploration revealed a markedly dilated gallbladder and CBD measuring approximately 10 × 7 cm, as well as hepatic ducts with an average diameter of 8 mm (Figure 4). The anatomical distinction between the gallbladder, cystic duct, and the CBD was no longer discernible. Additionally, the junction between the CBD and the duodenum at the level of the major duodenal papilla was severely dilated (Figure 5).

Both the gallbladder and CBD were filled with yellow exudate (a mixture of pus and bile), which began to leak from the liver upon initiation of dissection (Figure 6). Severe inflammation of the right arm of the pancreas was noted, along with dilation of the main pancreatic duct, which also contained yellow exudate. The stomach was empty and compressed, likely due to the mass effect exerted by the dilated gallbladder and CBD.

Following abdominal inspection, biopsies and bacteriological samples were collected from the liver, an enlarged mesenteric lymph node, the duodenum, and the right pancreatic limb. The gallbladder was carefully dissected from its fossa, and a Foley catheter was inserted into the CBD to allow draining of its content. The catheter was left in situ and functioned as a choledochostomy tube to allow biliary diversion and support tissue healing (Figure 7). A duodenotomy was performed to catheterize the CBD, which proved impossible due to complete irreversible obstruction of the major duodenal papilla. Despite the occlusion of the CBD, the main pancreatic duct remained patent and could be flushed. A cholecystectomy was performed, and the opening to the CBD was closed using polydioxanone 4-0 in a double-layer continuous suture pattern (appositional followed by Cushing) (Figure 8).

A jejunal feeding tube was introduced through the esophageal tube and advanced through the duodenotomy into the jejunum. A choledochoenterostomy was performed by anastomosing the dilated CBD to the jejunum using polydioxanone 4-0 in a single continuous suture pattern (Figure 9). After changing gloves and instruments, the abdominal cavity was lavaged. A Jackson–Pratt-like abdominal drain was placed, and routine abdominal closure was performed. The procedure was performed by an experienced boarded surgeon (SV).

### 2.12. Bacteriology and Histopathology

Hemolytic Escherichia coli was still present in the bile and remained sensitive to all commonly used antibiotics (Thiogluconate Enrichment broth, 5% sheep blood agar, and aerobe and anaerobe incubation at 37 °C; minimal inhibitory concentration (MIC) values ranged between 0.25 and 20; MIC value for Ampicillin at 2, for Marbofloxacin 0.5). Histopathological findings of the liver, bile, pancreas, and duodenum revealed a chronic-active, severe neutrophilic and lymphoplasmacytic inflammation with areas of necrosis. The mesenteric lymph node showed hyperplasia.

Although a definitive underlying cause could not be determined, a congenital malformation of the CBD and major duodenal papilla is suspected.

### 2.13. Aftercare and Outcome

The cat’s general condition, appetite, and blood values showed marked improvement following surgery. In addition to the aforementioned antimicrobials, the cat received continuous rate infusion (CRI) of Metoclopramide (Paspertin, Viatris Pharma GmbH, Steinhausen, Switzerland) at 1 mg/kg/day IV, Ondansetron (Ondansetron Labatec, Labatec Pharma SA, Meyrin, Switzerland) 0.3 mg/kg IV TID, Buprenorphin (Bupaq P, Streuli Pharma, Uznach, Switzerland) 0.01 mg/kg IV TID, Vitamin K supplementation (Konakion, Cheplapharm, Greifswald, Germany) 2.5 mg/kg IV BID, S-Adenosylmethionin-Silybin (Denamarin S, Nutramax Laboratories Inc., Lancester, SC, USA) 1 pill PO SID, and Ursodeoxycholic acid (Ursochol, Zambon Ltd., Cadempino, Switzerland) 10 mg/kg PO SID. However, three days postoperatively, the cat developed hypotension and hypoglycemia, which were interpreted as signs of sepsis. Consequently, a third antibiotic, Metronidazole (Metronidazol Sintetica, Sintetica SA, Mendrisio, Switzerland) 15 mg/kg IV TID was initiated, alongside glucose supplementation and vasopressor therapy with Noradrenaline CRI (Noradrenalin Sintetica, Sintetica SA, Mendrisio, Switzerland) 0.1–0.5 ug/kg/min. Due to suspicion of critical illness-related corticosteroid insufficiency (CIRCI), a Hydrocortisone CRI (Solu-Cortef, Pfizer AG, Zürich, Switzerland) at 3.8 mg/kg/day was added. Initially, the cat responded well to the therapy, but six days after surgery, clinical deterioration recurred with renewed hypotension and hypoglycemia. Additionally, the cat developed severe hypoalbuminemia (15 g/L), prompting administration of human albumin transfusion. Under this treatment, albumin levels improved significantly to 27.1 g/L, blood pressure stabilized, and Hydrocortisone could be gradually tapered. Despite this improvement, the cat died unexpectedly 10 days postoperatively. The last measured vital parameters, as well as the blood pressure, glucose levels, and blood work, were within normal limits. Necropsy was declined by the owners.

## 3. Discussion

The first human choledochoenterostomy procedures in the literature were described by Riedel in 1888, with the first successful case performed by Sprengel in a woman in 1891 [5,6]. Although choledochoenterostomy is extremely rarely performed in veterinary medicine, and has not previously been reported in cats, the earliest experimental trials for human applications were conducted in dogs [6]. Today, choledochoenterostomy is a routine surgical intervention in the management of hepatobiliary disorders in humans and is increasingly performed using minimally invasive laparoscopic techniques [7]. In veterinary medicine, the procedure has remained largely theoretical, primarily due to the small diameter of the CBD in small animals, which is only 2–4 mm in cats compared to 4–6 mm in dogs and 8 cm in length and 7 mm in diameter in humans [1,2,4,8]. In general, biliary surgery is more commonly performed in dogs than in cats due to a relatively larger extrahepatic biliary tract, which facilitates surgical access and anastomosis [1,4]. The small luminal diameter in cats increases the technical difficulty of surgery and heightens the risk of postoperative complications such as leakage, strictures, or obstruction due to minor misalignments or suture reaction [1,2,4].

The case described exemplifies a not-yet-documented instance of choledochoenterostomy in a feline patient, highlighting the exceptional nature of such an intervention due to an irreversible obstruction of the duodenal papilla and a severely altered gallbladder, which had to be excised. Due to the irreversible complete obstruction of the major duodenal papilla, attempted catheterization of the CBD was not successful, making choledochal stenting impossible. In this instance, the feasibility of a choledochoenterostomy was rendered possible only by an extreme pathological dilation of the CBD due to chronic obstruction and ascending cholangiohepatitis. Under normal anatomical conditions in cats, such a procedure would likely be technically unfeasible and contraindicated due to the minute size of the duct and the high risk of dehiscence or occlusion [9].

The postoperative period was characterized by initial improvement followed by rapid deterioration due to suspected septic complications and CIRCI. Furthermore, a thromboembolic complication cannot be ruled out, and sepsis, as well as CIRCI, may have led to liver or even multiorgan failure. Stabilization with vasopressors and corticosteroids, as well as nutritional and hepatic support, likely delayed decompensation. The key challenges were managing septic complications and critical illness-induced endocrinopathies, monitoring fluid balance and albumin levels, especially with ongoing inflammatory losses and possible multiorgan failure.

Despite temporary recovery, the severity of the initial disease, the complexity of the surgical correction, and secondary complications like CIRCI and hypoalbuminemia significantly increased the mortality risk. Surgical methods for biliary diversion have been associated with high perioperative morbidity and mortality rates and short overall survival times in cats [10,11,12]. Causes for the high perioperative morbidity and mortality rates in cats undergoing biliary diversion surgery have not yet been fully identified. Buote et al. [13] suggest that in cats undergoing surgery for biliary diversion, the underlying cause of the obstruction is significantly associated with the outcome, whereas neoplasia has a significantly shorter median survival time than with cats with chronic inflammatory disease. Since the inflammatory disease processes are usually chronic and progressive, affected cats can eventually develop signs of hepatobiliary, pancreatic, or intestinal failure even though bile flow is restored [13]. Refractory hypotension might be due to poor vascular responsiveness because of systemic endotoxemia or increased production of the vasodilatory compound nitric oxide or may be secondary to vagal effects associated with manipulation of the biliary tract [13,14].

The biliduodenal sphincter (sphincter of Oddi) serves as a critical physiological barrier, protecting the biliary system from retrograde gastrointestinal pressure fluctuations and ascending infections [15]. In this case, no evidence of suture dehiscence or worsening of the existing biliary congestion was observed postoperatively. However, given the absence of this natural barrier following choledochoenterostomy, the progression of cholangiohepatitis may have been exacerbated by ascending infection. Since a necropsy was not performed, the precise cause of death remains unknown.

Overall, this case contributes valuable insight into the rarely documented field of feline biliary surgery, reinforcing the idea that while certain techniques may be standard in canine or human medicine, their application in feline patients must be approached with caution, reserved for cases of extreme pathological adaptation and performed by experienced surgical teams. Furthermore, the postoperative care and poor prognosis in this disease process should not be neglected.

## 4. Conclusions

This case report presents the first documented choledochoenterostomy in a feline patient, performed in response to an irreversible obstruction of the major duodenal papilla with secondary chronic bacterial cholangiohepatitis and severe biliary tract dilation. Despite initial postoperative improvement, the cat succumbed to complications likely related to sepsis, hypoalbuminemia, critical illness-related corticosteroid insufficiency, and possible multiorgan failure 10 days after surgery.

This case highlights the importance of adequate preoperative imaging monitoring, meticulous surgical planning, and intensive peri- as well as postoperative monitoring and care.

It cannot be overemphasized that, while choledochoenterostomy may be a viable salvage option in rare, extreme cases, its use in feline medicine should remain highly selective. Moreover, the poor long-term prognosis—even in the context of a technically successful intervention—underscores the critical need for early diagnosis, robust postoperative care, and tempered expectations regarding outcome.

This report contributes to the limited knowledge on advanced biliary surgery in cats and may serve as a reference for similar future cases.

## Figures and Tables

**Figure 1 animals-15-02634-f001:**
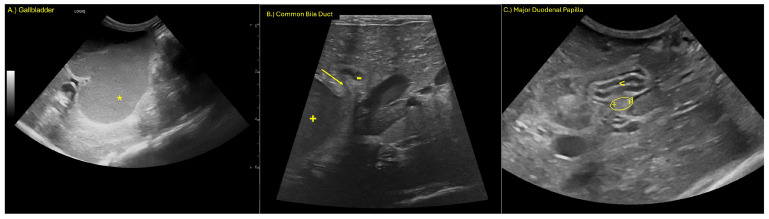
(**A**) Marked dilation of the gallbladder (*), (**B**) marked dilation of the CBD (+) with luminal tapering (arrow) and mild distension of the pancreatic duct (-), and (**C**) major duodenal papilla (oval) entering the duodenum (<).

**Figure 2 animals-15-02634-f002:**
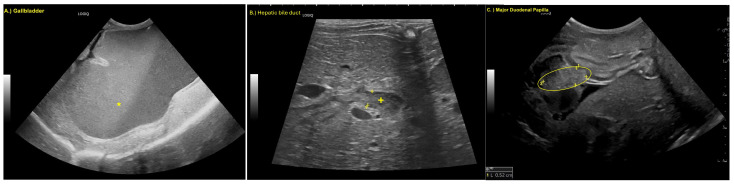
(**A**) Marked dilation of the gallbladder (*), (**B**) dilated hepatic bile duct, (+, boarders marked with 1 and +) and (**C**) thickened major duodenal papilla (oval, boarders of papilla marked with +, 1 and 2).

**Figure 3 animals-15-02634-f003:**
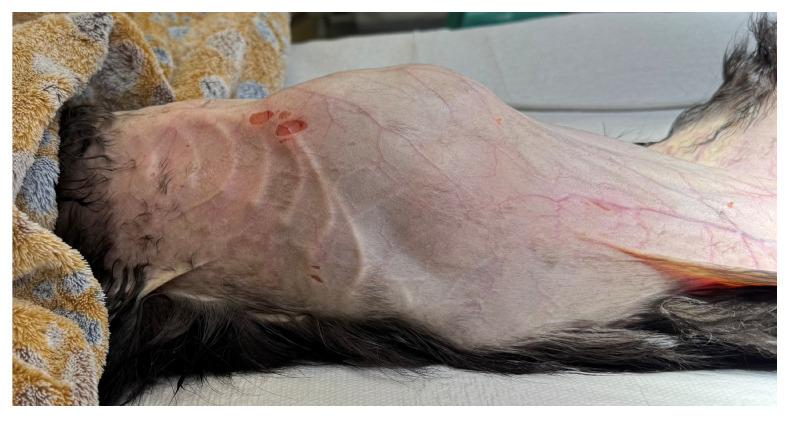
Enlargement of the cranial abdomen. Skin lesions due to clipping.

**Figure 4 animals-15-02634-f004:**
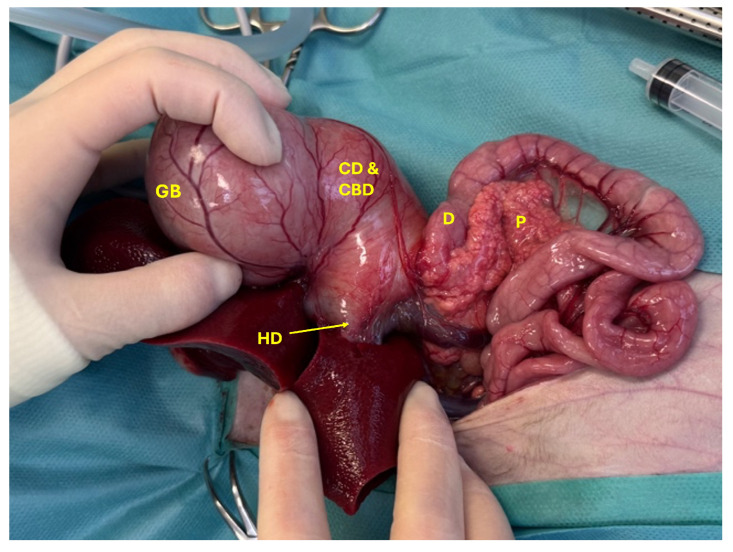
Dilated gallbladder (GB) with enlarged cystic and common bile duct (CD and CBD) entering into the duodenum (D), severely enlarged hepatic duct (HD), and severely inflamed pancreas (P).

**Figure 5 animals-15-02634-f005:**
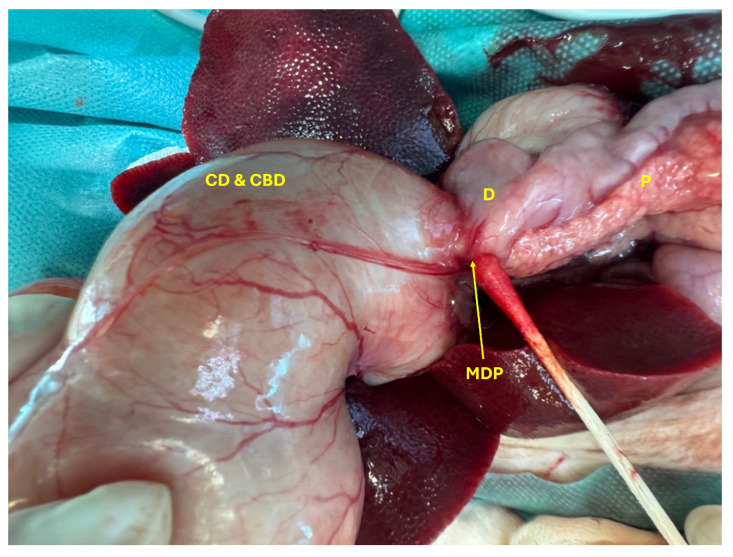
Severe dilation of the junction between the CD and CBD and the duodenum (D) at the major duodenal papilla (MDP) and an inflamed pancreas (P).

**Figure 6 animals-15-02634-f006:**
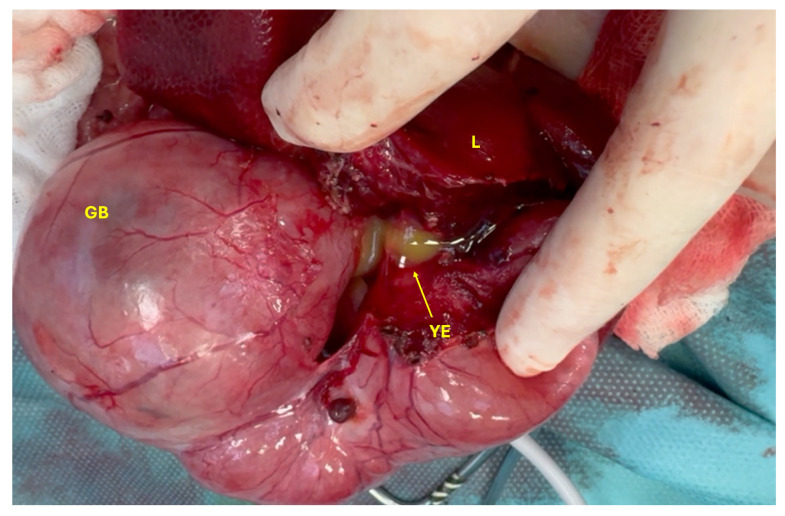
Yellow exudate (YE) from the gallbladder (GB) and liver (L).

**Figure 7 animals-15-02634-f007:**
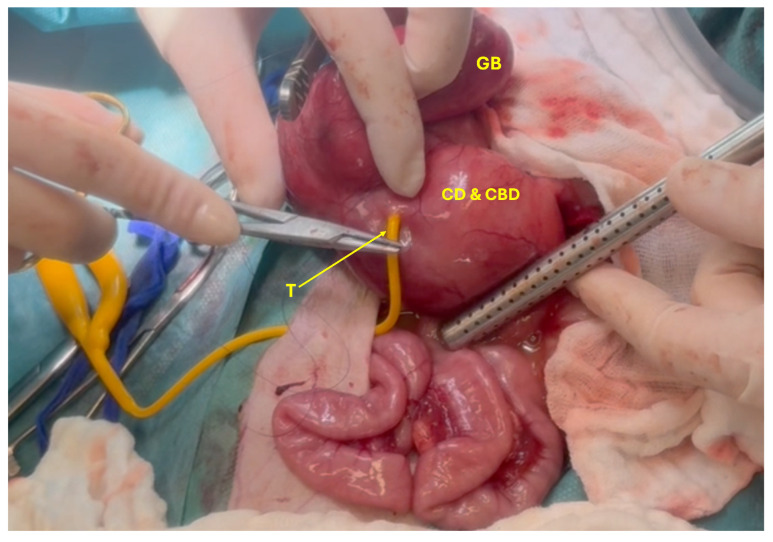
Placing of choledochostomy tube (T) in the cystic and common bile duct (CD and CBD) to empty CD, CBD and Gallbladder (GB).

**Figure 8 animals-15-02634-f008:**
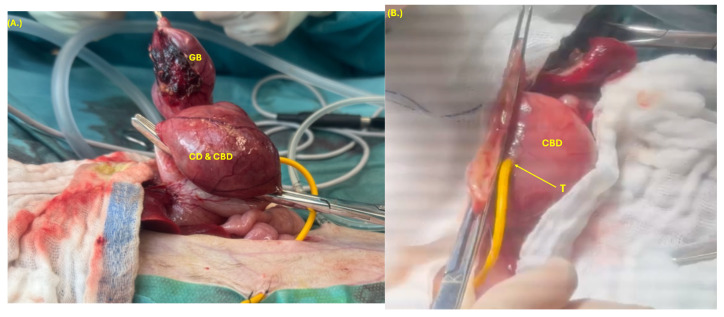
Cholecystectomy (**A**) of the gallbladder (GB), cystic duct (CD), and part of the dilated common bile duct (CBD) with a two-layer suture closure (**B**) of the CBD after cholecystectomy, and choledochostomy tube (T) in the dilated CBD.

**Figure 9 animals-15-02634-f009:**
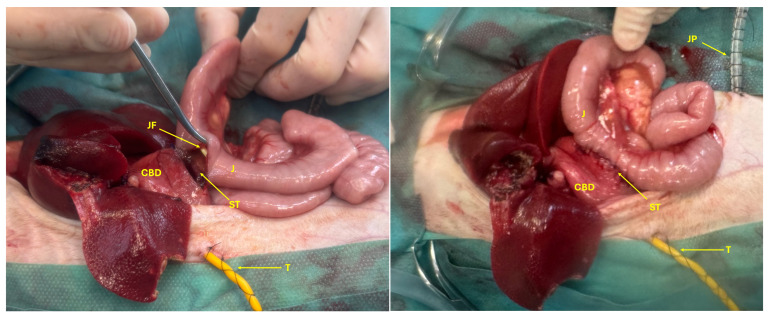
Choledochoenterostomy (ST) between the common bile duct (CBD) and jejunum (J) with the jejunal feeding tube (JF), choledochostomy tube (T), and the abdominal drain (JP).

**Table 1 animals-15-02634-t001:** Chemistry panel 27.11.

Chemistry Panel	27.11.24		
Parameter	Value	Reference	Unit
Sodium	**144**	147.0–157.0	mmol/L
Potassium	4.11	3.26–5.11	mmol/L
Chloride	113	113.0–123.0	mmol/L
Calcium	**2.3**	2.34–2.88	mmol/L
Phosphorus	1.05	0.77–1.89	mmol/L
Glucose	3.78	3.22–9.86	mmol/L
Cholesterine	2.58	1.95–7.18	mmol/L
Triglycerides	0.55	0.23–1.45	mmol/L
Total Protein	**90.8**	63.0–83.2	g/L
Albumin	**25.4**	30.0–43.3	g/L
Globuline	**65.4**	25.8–50.6	g/L
Urea	10.1	6.5–13.5	mmol/L
Creatinine	**43**	64.0–163.0	umol/L
Bilirubin	**3.7**	0.0–2.2	umol/L
Alanine Aminotransferase (ALT)	**300**	25.0–87.0	U/L
Alkaline Phosphatase (AP)	56	13.0–69.0	U/L
Asparate Aminotransferase (AST)	**70**	12.0–40.0	U/L
Creatinine Kinase	96	68.0–555.0	U/L
Gamma Glutamyl Transferase	**17**	<1	U/L
Glutamate Dehydrogenase	**44**	0.0–5.0	U/L
Lipase	**60**	8.0–39.0	U/L
Serum Amyloid A (SAA)	4.6	<5.2	mg/L

**Table 2 animals-15-02634-t002:** Chemistry panel 18.12.2024.

Chemistry Panel	18.12.24		
Parameter	Value	Reference	Unit
Sodium	**146**	147.0–157.0	mmol/L
Potassium	3.67	3.26–5.11	mmol/L
Chloride	114	113.0–123.0	mmol/L
Calcium	**2.17**	2.34–2.88	mmol/L
Phosphorus	0.82	0.77–1.89	mmol/L
Glucose	3.89	3.22–9.86	mmol/L
Cholesterine	3.23	1.95–7.18	mmol/L
Triglycerides	0.6	0.23–1.45	mmol/L
Total Protein	82.4	63.0–83.2	g/L
Albumin	**23.7**	30.0–43.3	g/L
Globuline	**58.7**	25.8–50.6	g/L
Urea	7.9	6.5–13.5	mmol/L
Creatinine	**46**	64.0–163.0	umol/L
Bilirubin	**29.2**	0.0–2.2	umol/L
Alanine Aminotransferase (ALT)	**1140**	25.0–87.0	U/L
Alkaline Phosphatase (AP)	**158**	13.0–69.0	U/L
Asparate Aminotransferase (AST)	**276**	12.0–40.0	U/L
Creatinine Kinase	128	68.0–555.0	U/L
Gamma Glutamyl Transferase	**22**	<1	U/L
Glutamate Dehydrogenase	**245**	0.0–5.0	U/L
Lipase	**111**	8.0–39.0	U/L
Serum Amyloid A (SAA)	**30.4**	<5.2	mg/L

**Table 3 animals-15-02634-t003:** Full coagulation profile 18.12.24.

Coagulation Profile	18.12.24		
Parameter	Value	Reference	Unit
PT	13	10.5–13.3	s
PTT	**66.7**	11.9–16.2	s
Fibrinogen	223	100–300	mg/dL
D-Dimere	**0.75**	<0.45	ug/mL

## Data Availability

The data presented in this study are available on request from the corresponding author upon reasonable request.

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
