# Peer review of "First Case Report of Choledochoenterostomy in a Cat with Biliary Obstruction Due to Cholangiohepatitis and Papillary Stenosis"

_animals, 2025, doi:10.3390/ani15172634_

Round 1

Reviewer 1 Report

Comments and Suggestions for Authors

Dear authors,

It was a pleasure reviewing your manuscript titled “First Report of Choledochoenterostomy in a Cat with Biliary Obstruction due to Cholangiohepatitis and Papillary Stenosis”. This case report represents an original and novel contribution, documenting for the first time a choledochoenterostomy in a feline patient with biliary obstruction. The manuscript is clinically relevant and well-structured, but requires important revisions before being considered for publication.

  • Firstly, I recommend that authors consider the standards proposed by the CARE guidelines for clinical cases, as it would help improve the clarity, comprehensiveness, and transparency of the manuscript (https://www.care-statement.org/).
  • Was there informed consent for the entire procedure from the owner? It should be explicitly stated that it was signed.
  • I suggest putting the chemical panel in a table, along with the reference values (those considered by the laboratory with which the analysis was performed) to facilitate its interpretation.
  • It is not clearly described whether there was coagulation assessment prior to surgery (only aPTT is mentioned)
  • The culture method and minimum inhibitory concentrations of antibiotics are not specified. Please explain this section further in detail.
  • The discussion does not sufficiently address why biliary stenting or less invasive techniques were not attempted beforehand. Furthermore, it does not analyze whether the death could have been due to thromboembolic complications or multiorgan failure (hypotheses could be raised without the necropsy).
  • Based on all of the above, the conclusions section should be modified, highlighting the limitations and expressing a more cautious approach in suggesting the feasibility of the procedure. Furthermore, practical clinical recommendations could be incorporated, such as the importance of early diagnosis or the risks inherent in biliary surgery in felines

Other suggestions.

  • The references section should be carefully reviewed to ensure it meets the journal's standards. For example, references 5 and 6 have mistakes.
  • The abbreviations section is incomplete. Please include all abbreviations contained in the text (PU/PD, PO, IV, ALAT, AST…..)
  • Review the references to figures and tables in the text (for example, Figures 2 and 3 are unreferenced)

Best regards.

Author Response

Dear Reviewer,

Thank you very much for your time, knowledge and valuable inputs to improve the quality of our manuscript.

The comments are addressed as following:

Firstly, I recommend that authors consider the standards proposed by the CARE guidelines for clinical cases, as it would help improve the clarity, comprehensiveness, and transparency of the manuscript (https://www.care-statement.org/).

  • Thank you very much for the comment and the suggestion of the CARE guidelines. We have adjusted the manuscript accordingly, integrating the failing information within the original paper structure.

Was there informed consent for the entire procedure from the owner? It should be explicitly stated that it was signed.

  • Thank you for this important comment. There was an informed consent signed by the owners, it is now mentioned in line 84-85 as well as in line 354-355. Furthermore, a copy of the signed informed consent was sent to the editor.

I suggest putting the chemical panel in a table, along with the reference values (those considered by the laboratory with which the analysis was performed) to facilitate its interpretation.

  • Thank you for the suggestion. The chemical panel is now visible in line 91-105 and 156 – 169.

It is not clearly described whether there was coagulation assessment prior to surgery (only aPTT is mentioned).

  • Thank you for raising this important point. There was a full coagulation assessment done before surgery, the complete coagulation panel is now included as a table in line 172.

The culture method and minimum inhibitory concentrations of antibiotics are not specified. Please explain this section further in detail.

  • Thank you for this insightful comment and the opportunity to clarify our methodology. We now provided a brief summary of the culture method and the relevant MIC for the antibiotics used available in line 127-128, 130 – 131 and 237-239.
    Specifically, we used for the Bile Thiogluconate Enrichment broth, 5% Sheep blood agar (aerobe and anaerobic incubation), brolac agar (aerobe) and for Salmonella clarification Tetrathionate Enrichment and selective agars (SALM ChromAgar® and Brilliant Green Agar). Temperature for all the growth conditions are 37°C. All the agars are ready to use agars bought from Oxoid Company. For abdominal effusion and liver swab we used Blood agar (aerobe, anaerobic), brolac agar and Thiogluconate enrichment broth. The MIC values vary for every antibiotic, the relevant ones are included in the manuscript. Since the very detailed description would go beyond the scope of the manuscript, we decided to summarize it.

The discussion does not sufficiently address why biliary stenting or less invasive techniques were not attempted beforehand. Furthermore, it does not analyze whether the death could have been due to thromboembolic complications or multiorgan failure (hypotheses could be raised without the necropsy).

  • Thank you for raising these important points. As described in line 214-216 catheterization of the CBD was attempted but not feasible due to irreversible obstruction of the major duodenal papilla. We have now expanded the discussion to explain the clinical decision-making process in line 288-290.
  • In addition, we appreciate the suggestion to consider potential causes of death, such as thromboembolic complications or multiorgan failure. We have included the hypothesis in line 296-298.

Based on all of the above, the conclusions section should be modified, highlighting the limitations and expressing a more cautious approach in suggesting the feasibility of the procedure. Furthermore, practical clinical recommendations could be incorporated, such as the importance of early diagnosis or the risks inherent in biliary surgery in felines.

  • Thank you very much for your valuable input. We have added a sentence and modified the wording according to your comment and suggestions in line 338-341 and 344.

Other suggestions: The references section should be carefully reviewed to ensure it meets the journal's standards. For example, references 5 and 6 have mistakes; The abbreviations section is incomplete. Please include all abbreviations contained in the text (PU/PD, PO, IV, ALAT, AST…..); Review the references to figures and tables in the text (for example, Figures 2 and 3 are unreferenced)

  • Your suggestions are very much appreciated to improve the quality of our manuscript. The references are reviewed and corrected, the abbreviations section is supplemented and the figures are revised and referenced.

All the changes in the manuscript are highlighted in yellow.

Best Regards,

The Authors.

Reviewer 2 Report

Comments and Suggestions for Authors

The paper is interesting and presents valuable clinical data that will be useful for veterinary practitioners. It is well written, and with minor revisions it could become a cornerstone for new therapies in feline gallbladder disorders. I have a few minor comments listed below:

Please indicate previous treatments and drugs used previously in the first-opinion practice.

What was the cat’s status regarding FeLV and FIV? Was it checked for FIP?

2.4: Please provide ultrasound (US) images.

Fig. 1: Please indicate the main findings on the images.

Fig. 2: Please comment on the visible skin lesions.

Figs. 3–8: Please label all visible anatomical structures on the images; you may use standard abbreviations (e.g., GB, CD, CBD, etc.).

Author Response

Dear Reviewer,

Thank you very much for your time, knowledge and valuable inputs to improve the quality of our manuscript.

The comments are addressed as following:

Please indicate previous treatments and drugs used previously in the first-opinion practice.

What was the cat’s status regarding FeLV and FIV? Was it checked for FIP?

  • Thank you very much for your valuable question. The previous treatments and drugs of the first opinion practice are now included in line 77-85. The FelV/FIV test by the first opinion practice was negative and now also included in the manuscript. The cat was negatively checked for FIP one year prior to this episode of illness, but since we did not have any suspicion of FIP the test was not repeated.

2.4: Please provide ultrasound (US) images.

-      Thank you for raising this important point. The US images are now included with an indication of the main findings.

Fig. 1: Please indicate the main findings on the images.

  • Your suggestion is very much appreciated. The main findings are now indicated and visible in all the figures.

Fig. 2: Please comment on the visible skin lesions.

  • Thank your for your insightful comment. The skin lesions occurred during clipping, it’s now stated in the figure description in line 198.

Figs. 3–8: Please label all visible anatomical structures on the images; you may use standard abbreviations (e.g., GB, CD, CBD, etc.).

  • Thank you for this very important comment. The anatomical structures are now labeled in every figure.

All the changes in the manuscript are highlighted in yellow.

Best Regards,

The Authors.

Round 2

Reviewer 1 Report

Comments and Suggestions for Authors

Dear authors,

The manuscript is now much more robust, clear, and transparent. You have addressed the main concerns.

Minor new suggestions:

  • Explicitly indicate compliance with the CARE guidelines, as the structure already does, and you should add the CARE checklist as supplementary material.
  • You should also review the references section again, as it does not meet the journal's standards.

Best regards.

Author Response

Dear Reviewer, 

Thank you for your thoughtful and constructive feedback. We greatly appreciate your positive comments regarding the improvements made to the manuscript. 

Thank you as well for your minor suggestions, they're addressed as following:

  • The compliance with the CARE guidelines in now indicated in the acknowledgments and the CARE checklist is added in the supplementary material (in the zip folder of the figures since the system does not allow me to upload supplementary material).
  • The reference section is carefully reviewed and match the journals standards of Author 1, A.B.; Author 2, C.D. Title of the article. Abbreviated Journal Name YearVolume, page range. for Journal Articles and Author 1, A.; Author 2, B. Book Title, 3rd ed.; Publisher: Publisher Location, Country, Year; pp. 154–196. for Books as well as Author 1, A.; Author 2, B. Title of the chapter. In Book Title, 2nd ed.; Editor 1, A., Editor 2, B., Eds.; Publisher: Publisher Location, Country, Year; Volume 3, pp. 154–196. for Book Chapters. Furthermore the DOI is included as stated on the website.

The two sections with changes are highlighted in yellow.

Best Regards,
The Authors.